# A New Short-Circuit Current Calculation and Fault Analysis Method Suitable for Doubly Fed Wind Farm Groups

**DOI:** 10.3390/s23208372

**Published:** 2023-10-10

**Authors:** Jun Yin, Weichen Qian, Xiaobo Huang

**Affiliations:** School of Electrical Engineering, North China University of Water Resources and Electric Power, Zhengzhou 450045, China

**Keywords:** doubly fed wind farm group, LVRT, short-circuit current, fault analysis

## Abstract

The transient characteristics of wind farms in groups are quite different; in addition, there is a strong coupling between the wind farms and the grid, and these factors make the fault analysis of the grid with wind farm groups complicated. In order to solve this problem, a mathematical model of the converter is established based on the input-output external characteristics of the converter, and a transient model of a doubly fed wind turbine (DFIG) is presented considering the influence of the low-voltage ride-through control (LVRT) of the converter, and the effect mechanism of the LVRT strategy on the short-circuit current is analyzed. Finally, a short-circuit current calculation model of a doubly fed wind turbine with low-voltage crossing control is established. The interaction mechanism between wind farms during the fault is analyzed, and a short-circuit current calculation method of doubly fed wind farm groups is proposed. RTDS is used to verify the accuracy of the proposed short-circuit current calculation method for doubly fed field groups. On this basis, a method of power grid fault analysis after doubly fed field group access is discussed and analyzed.

## 1. Introduction

Due to the characteristics of the regional distribution of wind resources in China, wind power generation is often built in two forms: cluster development and centralized grid connection [1]. At present, China is planning to build nine ten-million-kilowatt wind power bases [2]. If a wind farm group is connected to the grid, it is necessary to carry out corresponding research to meet the needs of large-scale grid connection of new energy [3].

Some field data show that in some areas where wind farms are clustered, wind farms cause large fault currents [4]. Therefore, considering that the accuracy of short-circuit current calculation also has a great influence on the relay protection action, it is necessary to propose a new method for calculating short-circuit current.

At present, the existing literature has focused on the problems caused by the integration of wind farm groups into the grid [5]. Some literature [6] analyzes the impact of large-capacity wind farms on the transient stability of the power grid [7].

Most of the existing research considers wind farm groups as equivalent to a wind turbine of equal capacity [8]. However, the centralized grid-connected wind farms are distributed according to the characteristics of wind resources, the actual distance between farms is relatively far, and the operating conditions of each wind farm are different, which makes their transient characteristics after fault greatly different [9]. In addition, there is a strong coupling relationship between each wind farm and the system [10]. Therefore, it is significant to propose a method to calculate the short-circuit current considering the coupling relationship between wind farm groups and the system [11].

In addition, in order to meet the requirement of providing a certain voltage support during a fault [12]. The converter of the DFIG cannot simply be blocked using crowbar during the fault period, and the rotor excitation needs to be adjusted through the LVRT control strategy to output reactive current [13]. In this case, the output short-circuit current of the DFIG is affected by the converter control characteristics [14]. Therefore, it is necessary to consider the low-voltage crossing control strategy and the control characteristics of the converter when calculating the short-circuit current [15].

In this paper, a transient model of doubly fed wind turbines is proposed, which takes into account the influence of the low-voltage crossing control strategy [16]. Considering the transient model and the influence of converter control characteristics, a DFIG short-circuit current calculation model is established [17]. The coupling relationship between the short-circuit current of the wind farm and the voltage of the grid node is further given, and the interaction mechanism of the short-circuit current within the wind farm groups and between the wind farm and the grid system is analyzed; On this basis, this paper proposes a calculation method of short-circuit current suitable for wind farm group access [18]. Secondly, in order to verify the accuracy of the proposed short-circuit current calculation method, a doubly fed field group experimental platform is established through RTDS. Finally, on the basis of further analysis of a doubly fed wind farm equivalent model, this paper proposes a fault analysis method for a group of doubly fed wind farms connected to the electricity grid.

## 2. A Short-Circuit Current Calculation Method Considering Low-Voltage Crossing Control

### 2.1. DFIG Transient Model Considering the Control Characteristics of the Converter

In previous studies, it was believed that after a fault occurred, crowbar was switched on and the rotor converter was blocked [19]. According to wind power grid-connected specifications, the converter needs to emit a reactive current to support grid voltage recovery during low-voltage traversal [20]. In addition, due to the transient process of a DFIG, there will be a certain effect because of the rotor- side converter atresia. Therefore, the converter input and output characteristics should also be considered in the analysis of the electromagnetic transient process.

It has been shown that power imbalances between converters can be suppressed by using the chopper circuit in the DC bus [21,22]. Thus, the DC bus voltage during the fault period is guaranteed to be near the rated value [23]. Therefore, it is assumed in this paper that DC voltage u_dc_ can remain stable after failure, DC voltage u_dc_ can be obtained using the rotor-side converter, and the excitation voltage u_r_ can be adjusted through the converter duty ratio. Then, the rotor- side excitation voltage u_r_ can be expressed as follows:(1)$$u_{r}=\frac{A}{n}K_{\mathrm{dc}}u_{\mathrm{dc}}-i_{\mathrm{dc}}X_{\mathrm{rsc}}-\Delta u_{\mathrm{rsc}}$$
where ***A***, ***n***, and ***K_dc_*** are the current inverting coefficient, ratio, and duty cycle, respectively; ***i_dc_***, ***X_rsc_***, and Δ***u_rsc_*** refer to the DC bus current, equivalent arcing reactance of converter, and IGBT voltage drop, respectively.

In the transient process, the magnetic saturation phenomenon is not considered and the rotation speed is assumed to be constant, then motor convention is adopted [24]. The DFIG space vector model under a synchronous rotation coordinate system is as follows [25,26]:(2)$$\left[\begin{array}{c} u_{s}\\ u_{r} \end{array}\right]=\left[\begin{array}{cc} R_{s} & 0\\ 0 & R_{r} \end{array}\right]\left[\begin{array}{c} i_{s}\\ i_{r} \end{array}\right]+\left[\begin{array}{c} \frac{d\psi_{s}}{dt}\\ \frac{d\psi_{r}}{dt} \end{array}\right]+\left[\begin{array}{cc} j\omega_{s} & 0\\ 0 & j\omega_{s-r} \end{array}\right]\left[\begin{array}{c} \psi_{s}\\ \psi_{r} \end{array}\right]$$
(3)$$\left[\begin{array}{c} \psi_{s}\\ \psi_{r} \end{array}\right]=\left[\begin{array}{cc} L_{s} & L_{m}\\ L_{m} & L_{r} \end{array}\right]\left[\begin{array}{c} i_{s}\\ i_{r} \end{array}\right]$$

Equation (2) gives the relationship between stator voltage ***u_s_*** and rotor voltage ***u_r_*** after converting the stator side; Equation (3) gives the relationship between the DQ-axis flux ***ψ_s_*** and ***ψ_r_*** and the stator and rotor currents.

The synthesis of Equations (1)–(3) is shown in Figure 1.

Here ***E_g_*** is the equivalent potential at the grid side, ***Z_1L_*** is the equivalent impedance from the system to the fault point, the equivalent impedance from the doubly fed wind turbine to the fault point is ***Z_2L_***, the transition impedance is ***Z_f_***, the filter resistance and reactance of the grid-measured converter are ***R_g_*** and ***L_g_***, and the voltage at the end of the DFIG machine is us.

### 2.2. Study on Short-Circuit Current Variation Mechanism Considering Low-Voltage Crossing Control

As shown in Figure 1, the DFIG rotor converter can complete the low-voltage crossing by controlling the rotor excitation voltage ***u_r_*** during the grid voltage drop. Therefore, the effect of the excitation voltage ***u_r_*** cannot be ignored.

The rotor voltage equation can be obtained by combining Equations (2) and (3) to eliminate the rotor current:(4)$$u_{r}=\frac{L_{m}}{L_{s}}\frac{d\psi_{s}}{dt}+j\omega_{s-r}\frac{L_{m}}{L_{s}}\frac{d\psi_{s}}{dt}+(\frac{L_{r}L_{s}-L_{m}^{2}}{L_{s}})\frac{di_{r}}{dt}+R_{r}i_{r}+j\omega_{s-r}(\frac{L_{r}L_{s}-L_{m}^{2}}{L_{s}})i_{r}$$

Converted to the rotor rotating coordinate system, Formula (4) is simplified as follows:(5)$$u_{r}^{r}=\frac{L_{m}}{L_{s}}\frac{d\psi_{s}^{r}}{dt}+R_{r}i_{r}^{r}+(\frac{L_{r}L_{s}-L_{m}^{2}}{L_{s}})\frac{di_{r}^{r}}{dt}$$

In the formula, $u_{r}^{r}$
$\psi_{s}^{r}$ and ***i_r_*** are the rotator voltage, the stator magnetic chain, and the rotator current in the rotating coordinate system.

Considering that the DC attenuation component of the short-circuit current of the DFIG group is closely related to the stator-side flux *d****Ψ**_s_*/***dt*, the relationship between the DC attenuation component and the stator-side flux is better discussed.This article does not consider the impact of the stator flux transient process with the rotor-side converter in the inertial time of IGBT. Thus, it can be assumed that ***K_dc_*** immediately changes to the corresponding modulation ratio after the fault occurs, and the rotor-side voltage is quickly adjusted to the reference value ***u***_r_, by the rotor converter. Formula (6) can be obtained by combining Formula (1) and Formula (5) as follows.
(6)$$L_{r}^{\prime }\frac{di_{r}^{r}}{dt}+R_{r}i_{r}^{r}-u_{r}^{r}=0$$
where, the rotor transient inductance $L_{r}^{\prime }$ is $L_{r}^{\prime }$ = *L*_r_ − $L_{m}^{2}$$L_{s}^{-1}$.

According to Equation (6), the following can be obtained:(7)$$i_{r}(t)=(i_{r0}^{\prime }-i_{r\_\mathrm{ref}})e^{-t/\tau_{r}}+i_{r\_\mathrm{ref}}$$

Considering the parameters of the DFIG, ***L_s_*** = ***L_sσ_*** + ***L_m_***, ***L_r_*** = ***L_rσ_*** + ***L_m_*** and ***L_m_*** ≥≥ ***L_sσ_***, ***L_m_*** ≥≥ ***L_rσ_***, then the stator and rotor currents can be expressed by eliminating Formula (3):(8)$$\left\{\begin{array}{l} i_{s}=\frac{L_{r}\psi_{s}-L_{m}\psi_{r}}{L_{s}L_{r}-L_{m}^{2}}\approx \frac{\psi_{s}-\psi_{r}}{L_{s}{}_{\sigma }+L_{r}{}_{\sigma }}\\ i_{r}=\frac{-L_{m}\psi_{s}+L_{s}\psi_{r}}{L_{s}L_{r}-L_{m}^{2}}\approx \frac{-(\psi_{s}-\psi_{r})}{L_{s}{}_{\sigma }+L_{r}{}_{\sigma }}\approx -i_{s} \end{array}\right.$$

Equation (8) reflects the opposite approximate magnitude of stator current and rotor current during the fault, so Equation (7) can be rewritten as follows:(9)$$i_{s}(t)=(i_{s0}^{\prime }-i_{s\_\mathrm{ref}}^{})e^{-t/\tau_{r}}+i_{s\_\mathrm{ref}}^{}$$
where ***i^′^_s0_*** and ***i_s_ref_*** are stator side short-circuit currents corresponding to the low-through control strategy at the initial and steady time of grid voltage sag, respectively. Equation (9) reflects that the variation rule of short-circuit current is related to the initial value and steady-state value of stator short-circuit current.

### 2.3. Short-Circuit Current Calculation Method of DFIG Group at the Initial Time of Grid Voltage Sag

The stator flux expression can be obtained from Equation (3):(10)$$\psi_{s}=\frac{L_{m}}{L_{r}}\psi_{r}+(\frac{L_{s}L_{r}-L_{m}^{2}}{L_{r}})i_{s}$$

The stator voltage equation can be obtained from Equations (2,10) as follows:(11)$$\begin{array}{l} \frac{Z_{f}E_{g}}{Z_{f}+Z_{1L}}=R_{s}i_{s}+j\omega_{s}\frac{L_{m}}{L_{r}}\psi_{r}+j\omega_{s}(\frac{L_{s}L_{r}-L_{m}^{2}}{L_{r}})i_{s}+\frac{d\psi_{s}}{dt}+i_{s}\frac{Z_{2L}(Z_{f}+Z_{1L})+Z_{f}Z_{1L}}{Z_{f}+Z_{1L}} \end{array}$$

In the initial stage of the fault, the stator flux d***ψ****_r_*/dt = 0 can be considered. Then, the initial moment of short-circuit current can be simplified into the following form (Equation (12)):(12)$$i_{s0}^{\prime }=(j\omega_{s}\frac{L_{m}}{L_{r}}\psi_{r0}-\frac{Z_{f}E_{g}}{Z_{f}+Z_{1L}})/-(R_{s}+X^{\prime }+\frac{Z_{2L}(Z_{f}+Z_{1L})+Z_{f}Z_{1L}}{Z_{f}+Z_{1L}})$$

According to Equation (12), only ***ψ***_r0_ is an unknown quantity in the fault current of the doubly fed unit at the initial time.

Considering that the rotor flux of the DFIG group cannot be mutated at the fault instant, before the fault, the output active power and reactive power of the DFIG group are as follows:(13)$$\left\{\begin{array}{l} P_{0}=\frac{3}{2}(u_{\mathrm{sq}}i_{\mathrm{sq}}+u_{\mathrm{sd}}i_{\mathrm{sd}})=\frac{3}{2}u_{\mathrm{sq}}i_{\mathrm{sq}}=\frac{3}{2}u_{s}i_{\mathrm{sq}}\\ Q_{0}=\frac{3}{2}(u_{\mathrm{sq}}i_{\mathrm{sd}}-u_{\mathrm{sd}}i_{\mathrm{sq}})=\frac{3}{2}u_{\mathrm{sq}}i_{\mathrm{sd}}=\frac{3}{2}u_{s}i_{\mathrm{sd}} \end{array}\right.$$

As the stator flux orientation control strategy is adopted on the rotor side of the DFIG group, ***ψ**_sd_*** = ***ψ**_s_*** = ***u**_s_***/***jω_s_***, ***ψ**_s_*** is the amplitude of the stator flux. Formula (3) can be rewritten as follows:(14)$$\left\{\begin{array}{l} \psi_{sd}=L_{s}i_{\mathrm{sd}}+L_{m}i_{\mathrm{rd}}=\psi_{s}=\frac{u_{s}}{j\omega_{s}}\\ \psi_{sq}=L_{s}i_{\mathrm{sq}}+L_{m}i_{\mathrm{rq}}=0\\ \psi_{\mathrm{rd}}=L_{m}i_{\mathrm{sd}}+L_{r}i_{\mathrm{rd}}\\ \psi_{\mathrm{rq}}=L_{m}i_{\mathrm{sq}}+L_{r}i_{\mathrm{rq}} \end{array}\right.$$

According to Equations (13) and (14), the relationship between rotor flux and DFIG output active power, reactive power, and pre-fault voltage at the initial time of fault can be expressed as follows:(15)$$\left\{\begin{array}{l} \psi_{\mathrm{rd0}}=\frac{2Q_{0}}{3u_{s}}(\frac{L_{m}^{2}-L_{r}L_{s}}{L_{m}})+\frac{L_{r}}{L_{m}}\frac{u_{s0}}{j\omega_{s}}\\ \psi_{\mathrm{rq0}}=\frac{2P_{0}}{3u_{s}}(\frac{L_{m}^{2}-L_{r}L_{s}}{L_{m}})\\ \psi_{r0}=\psi_{\mathrm{rd0}}+j\psi_{\mathrm{rq0}} \end{array}\right.$$

Assuming that the stator voltage ***u_s*0*_*** before the fault is rated, the active and reactive power depend on the setting of the working conditions before the network failure. The rotor flux ***ψ_r_*** is obtained from Equation (15) and the rotor flux ***ψ***_r_ is brought into Equation (12) to obtain the short circuit current ${i^{\prime }}_{so}$ at the initial moment of grid voltage drop.

### 2.4. Short-Circuit Current Calculation of DFIG Group at Fault Steady-State Time

Due to the low-voltage crossing period, the rotor-side converter needs to emit a reactive current to support the grid voltage recovery when the grid voltage drops [27,28,29]. When terminal voltage sag is detected after the fault, the rotor converter control strategy is adjusted to output reactive current in the low-voltage traverse control mode, and the rotor excitation current reference value is adjusted to output reactive power according to the wind power grid connection standard [30,31,32,33].

When the fault reaches the steady state, that is, d***ψ**_s_***/***dt*** = 0, the following can be obtained from Formula (2):(16)$$\frac{Z_{f}E_{g}}{Z_{f}+Z_{1L}}-i_{s}\frac{Z_{2L}(Z_{f}+Z_{1L})+Z_{f}Z_{1L}}{Z_{f}+Z_{1L}}=R_{s}i_{s}+j\omega_{s}\psi_{s}$$

According to Formulas (3,16), it can be obtained that:(17)$$\frac{Z_{f}E_{g}}{Z_{f}+Z_{1L}}-i_{s\_\mathrm{ref}}\frac{Z_{2L}(Z_{f}+Z_{1L})+Z_{f}Z_{1L}}{Z_{f}+Z_{1L}}=R_{s}i_{s\_\mathrm{ref}}+j\omega_{s}(L_{s}i_{s\_\mathrm{ref}}+L_{m}i_{r\_\mathrm{ref}})$$
where, ***i_r_ref_*** = ***i_rd_ref_*** + ***ji_rq_ref_***, ***i_rq_ref_*** and ***i_rd_ref_*** are the rotor active and reactive current reference values, respectively.
(18)$$i_{s\_\mathrm{ref}}=(-\omega_{s}L_{m}i_{\mathrm{rq}\_\mathrm{ref}}^{}+j\omega_{s}L_{m}i_{\mathrm{rd}\_\mathrm{ref}}^{}-\frac{Z_{f}E_{g}}{Z_{f}+Z_{1L}})/-(R_{s}+X+\frac{Z_{2L}(Z_{f}+Z_{1L})+Z_{f}Z_{1L}}{Z_{f}+Z_{1L}})$$
where, *X* = ***jω_s_L_s_***.

According to Equation (18), *i**_s_ref_*** is determined by ***i_rq_ref_***, ***i_rd_ref_***, ***R_s_***, ***X***, ***E_g_***, ***Z_1L_***, ***Z_2L_***, ***Z_f_***, ***L_mb_***, and ***ω_s_***, where only ***i_rd_ref_*** and ***i_rd_ref_*** are unknown quantities.

According to China’s grid-connected standard, the reference values of the field current after the fault can be determined as ***i_d_ref_*** and ***i_q_ref_***:(19)$$\left\{\begin{array}{l} i_{\mathrm{rd}\_\mathrm{ref}}^{}=f(u_{s})=(\frac{u_{s0}}{j\omega_{s}L_{m}}-\frac{2L_{s}Q_{0}}{3u_{s0}L_{m}})+K_{d}(0.9-u_{s})i_{\mathrm{rN}},(K_{d}\geq 1.5)\\ i_{\mathrm{rq}\_\mathrm{ref}}^{}=-\frac{2}{3}\frac{L_{s}P_{0}}{L_{m}u_{s0}},(0\leq i_{\mathrm{rq}\_\mathrm{ref}}^{}\leq \sqrt{i_{\mathrm{rmax}}^{2}-i_{\mathrm{rd}\_\mathrm{ref}}^{2}}) \end{array}\right.$$
where ***i_rN_*** and ***i_rmax_*** are the rated current and upper current of the rotor and; ***K_d_*** is the reactive current gain coefficient determined using the grid connection standard.

According to Equations (13) and (14), the rotor current at the initial time can be obtained by eliminating the stator current:(20)$$\left\{\begin{array}{l} i_{rd0}=\frac{u_{s0}}{j\omega_{s}L_{m}}-\frac{2L_{s}Q_{0}}{3u_{s0}L_{m}}\\ i_{\mathrm{rq}0}=-\frac{2}{3}\frac{L_{s}P_{0}}{L_{m}u_{s0}}\\ i_{r0}=i_{rd0}+ji_{\mathrm{rq}0} \end{array}\right.$$

Formula (20) is brought into Formula (19) to obtain the following:(21)$$\left\{\begin{array}{l} i_{\mathrm{rd}\_\mathrm{ref}}^{}=f(u_{s})=(\frac{u_{s0}}{j\omega_{s}L_{m}}-\frac{2L_{s}Q_{0}}{3u_{s0}L_{m}})+K_{d}(0.9-u_{s})i_{\mathrm{rN}},(K_{d}\geq 1.5)\\ i_{\mathrm{rq}\_\mathrm{ref}}^{}=-\frac{2}{3}\frac{L_{s}P_{0}}{L_{m}u_{s0}},(0\leq i_{\mathrm{rq}\_\mathrm{ref}}^{}\leq \sqrt{i_{\mathrm{rmax}}^{2}-i_{\mathrm{rd}\_\mathrm{ref}}^{2}}) \end{array}\right.$$

After the grid voltage drops, the rotor excitation current is adjusted according to Equation (21). Since the active power ***P_*0*_*** cannot be mutated after the fault. The reactive current emission is only related to the voltage sag degree. Considering the requirement of the grid-connected code to issue reactive current to support the voltage recovery of the grid, the rotor-side converter should first determine the reference value of reactive current and then determine the reference value of active current.

Equation (21) shows the relationship between the reference value of reactive current and the drop depth, as shown in Figure 2. The dashed line area represents the control dead zone.

In Figure 2, ***u_sN_*** and ***Δu_s_*** are the rated terminal voltage and the amount of terminal voltage drop, respectively; ***i_rN_*** and Δ***i_rd_ref_*** are the rated rotor excitation current and reactive current reference value change s, respectively.

Assuming ***X_m_*** = *j**ω_s_L_m_***, according to Equations (18) and (21), the short-circuit steady-state current is
(22)$$i_{s\_\mathrm{ref}}=\frac{\frac{2}{3}\frac{jX_{m}L_{s}P_{0}}{L_{m}u_{s0}}-X_{m}(\frac{u_{s0}}{X_{m}}-\frac{2L_{s}Q_{0}}{3u_{s0}L_{m}})-X_{m}K_{d}0.9i_{\mathrm{rN}}+\frac{Z_{f}E_{g}}{Z_{f}+Z_{1L}}}{R_{s}+X+\frac{Z_{2L}(Z_{f}+Z_{1L})+Z_{f}Z_{1L}}{Z_{f}+Z_{1L}}(1+X_{m}K_{d}i_{\mathrm{rN}})}$$

As can be seen from Equation (22), the short-circuit current ***i_s_ref_*** in the fault steady state is only related to the voltage at the initial moment, the electromotive force (***E_g_***) at the grid side, the line impedance parameter, and power ***u_s*0*_***, ***P_*0*_***, ***Q_*0*_***. Combining Equations (9), (12) and (22), we can obtain the short-circuit current value of the doubly fed wind turbine during the whole transient process.

## 3. Short-Circuit Current Calculation Method for Doubly Fed Field Group Access

Because of the different capacities and, operating conditions and the long distance between wind farms, their transient characteristics after failure are quite different. In addition, there is a strong coupling relationship between wind farms. Therefore, the research on DFIG wind farm groups cannot be simply based on the calculation method of the short-circuit current of a single DFIG wind turbine. It is necessary to analyze the fault interaction mechanism between each wind farm in the backcourt group. Furthermore, a short-circuit current calculation method suitable for a doubly fed field group connected to the power grid is proposed.

Without loss of generality, the actual power grid in a certain area shown in Figure 3 is taken as an example for specific analysis. Since the fans in each wind farm in this example are of the same model, their transient characteristics are essentially consistent during the fault. In this paper, each wind farm is equivalent to a doubly fed wind turbine with equal capacity instead.

Among them, ***E_g_*** is the system equivalent potential; ***Z_g_*** and ***Z_ln_*** are the system equivalent impedance and the impedance of the line. ***I_g_*** and ***I_DFIGi_*** are the short-circuit currents provided by the system and each doubly fed wind farm, respectively.

The rotor flux ***jω_s_L_m_***$L_{r}^{-1}$***ψ_r0_*** in Formula (12) at the initial time is equivalent to the electric potential $E_{DFIGi}^{\prime }$, Combined with the fault equivalent circuit, the initial short-circuit current ***I*^(0)^** of each doubly fed wind farm can be calculated.
(23)$${\left[\begin{array}{ccccc} E_{g} & E_{\mathrm{DFIG}1}^{\prime } & E_{\mathrm{DFIG}2}^{\prime } & E_{\mathrm{DFIG}3}^{\prime } & E_{\mathrm{DFIG}4}^{\prime } \end{array}\right]}^{T}=Z^{\prime }I_{}^{\left(0\right)}$$
where Z′ is the grid impedance matrix; and $I_{}^{\left(0\right)}={\left[\begin{array}{ccccc} I_{g} & I_{\mathrm{DFIG}1}^{\left(0\right)} & I_{\mathrm{DFIG}2}^{\left(0\right)} & I_{\mathrm{DFIG}3}^{\left(0\right)} & I_{\mathrm{DFIG}4}^{\left(0\right)} \end{array}\right]}^{T}$.

When the initial short-circuit current I^(0)^ of each wind farm is injected into the grid, it will affect the voltage of each node in the grid. The short-circuit current of each wind farm at the next moment can be obtained from the attenuation law of the node voltage through Equations (9) and (22). At the same time, this will also lead to changes in the voltage of each node of the grid, making the wind farm group and the grid interact with each other.

In order to accurately calculate the short-circuit current of the doubly fed wind farm group after being connected to the grid, in addition to considering the coupling relationship between the field group and the grid, the step size **Δ***t* is selected as 0.1 ms, and the short-circuit current of each wind farm in each fault period is calculated hourly.

Through the above operation solution, together with the grid impedance matrix, the voltage $U_{i}^{\left(1\right)}$ of each grid node in the first **Δ***t* can be obtained. Since the calculation step is short, it can be considered that the voltage in this time period is constant, and the voltage $U_{i}^{\left(1\right)}$ in this period is incorporated into Equation (22) to obtain $I_{s\_refi}^{\left(1\right)}$. The short-circuit current $I_{DFIGi}^{\left(2\right)}$ of each doubly fed wind farm at the end of the step is further obtained according to Equation (9).

According to the short-circuit current $I_{DFIGi}^{\left(1\right)}$ at the end of the first **Δ***t* period of each wind farm, combined with the grid impedance matrix, the system node voltage $U_{i}^{\left(2\right)}$ in the second Δt can be obtained. Formula (22) is incorporated to obtain the $I_{s\_refi}^{\left(2\right)}$ of the step, and further, the short-circuit current $I_{DFIGi}^{\left(2\right)}$ of each doubly fed wind farm at the end of the step is obtained according to Formula (9).

Similarly, the node voltage $U_{i}^{\left(k\right)}$ in the kth calculation step Δ*t* can be brought into Formulas (22) and (9) to obtain the short-circuit current $I_{DFIGi}^{\left(k\right)}$ of the doubly fed wind farm at the end of the step.and calculate each node voltage $U_{i}^{\left(k+1\right)}$ in the next Δ*t* time period.

When the calculation time is longer than the fault recovery time, that is, when *t* ≧ *t_a_*, the fault analysis ends, and the short-circuit current effective value of each DFIG wind farm is output, as shown in Figure 4 for the above calculation process.

When the power grid voltage asymmetrical drop occurs, PLL can be used to obtain the positive sequence voltage phase and amplitude, so that the active and reactive current reference values can be obtained according to Equation (21). Therefore, when an asymmetric fault occurs, the expression of the positive sequence current of the DFIG is the same as that of Formula (22), which conforms to the law of change of Formula (9).

According to the system circuit structure, a negative sequence equivalent circuit can be established. Further, combining the boundary conditions of asymmetric faults, a composite sequence network circuit is established. According to the above calculation method, the asymmetric short-circuit current of the doubly fed wind farm group can be solved.

## 4. Doubly Fed Wind Power Grid Group of Failure Analysis

When the DFIG adopts the low-voltage through-through control mode, the inverter can output reactive current support grid voltage support recovery and no longer latch, providing continuous excitation. During the fault, this can be thought of as DFIG excitation to produce continuous power frequency electric potential.

According to the above analysis, at the fault steady-state moment, a steady short-circuit current that can be made of type (18) is obtained. Let ***ω_s_L_m_i_rq_ref_
***+ j***ω_s_L_m_i_rd_ref_*** be the steady-state equivalent potential ***E_DFIGi_*** of the doubly fed wind turbine, then the DFIG equivalent circuit can be expressed in the form of ***E_DFIGi_*** and steady-state impedance ***Z_DFIGi_*** in series in the fault steady state.

In case of asymmetric fault, the expression of DFIG rotor current the same type (21). It can be seen that in the case of asymmetric faults, only positive sequence potentials still exist in the DFIG group.

According to the above analysis, when symmetrical and asymmetrical faults occur in the system, the doubly fed wind turbine only has the positive internal sequence potential of power frequency in the fault steady state. According to Equations (18) and (21), the steady-state internal potential of the doubly fed wind turbine can be obtained as follows.
(24)$$\left\{\begin{array}{l} i_{\mathrm{rd}\_\mathrm{ref}}^{}=f(u_{s})=(\frac{u_{s0}}{j\omega_{s}L_{m}}-\frac{2L_{s}Q_{0}}{3u_{s0}L_{m}})+K_{d}(0.9-u_{s})i_{\mathrm{rN}},(K_{d}\geq 1.5)\\ i_{\mathrm{rq}\_\mathrm{ref}}^{}=-\frac{2}{3}\frac{L_{s}P_{0}}{L_{m}u_{s0}},(0\leq i_{\mathrm{rq}\_\mathrm{ref}}^{}\leq \sqrt{i_{\mathrm{rmax}}^{2}-i_{\mathrm{rd}\_\mathrm{ref}}^{2}})\\ E_{\mathrm{DFIGi}}=-\omega_{s}L_{m}i_{\mathrm{rq}\_\mathrm{ref}}^{}+j\omega_{s}L_{m}i_{\mathrm{rd}\_\mathrm{ref}}^{}\\ Z_{\mathrm{DFIGi}}=R_{s\_\mathrm{DFIGi}}+X_{\mathrm{DFIGi}} \end{array}\right.$$

According to Equation (24), the DFIG cannot be treated as a constant voltage source before and after the fault. It is necessary to use the characteristics of the fault steady DFIG equivalent potential to analyze the interaction mechanism between the doubly fed farm group and power grid. It is necessary to propose power grid fault analysis method for after the doubly fed field group is connected.

Suppose that point A in Figure 3 has a two-phase short-circuit of AB, The positive sequence and negative sequence network diagram of the power grid fault is shown in Figure 5. Where, ***E_g_*** and ***Z_g_*** are the equivalent potential and equivalent impedance of the system, respectively; ***E_DFIGi_***_,_
***Z_DFIGi_***, and ***Z_ln_*** are the equivalent potential and steady-state impedance of each doubly fed wind farm and the equivalent impedance of the line ***L_n_***, respectively.

When the DFIG group is connected to the power grid, the system will become a form of multi-power supply. After the power grid fails, considering the mutual coupling between wind farms and the influence of the control strategy, there is a strong non-linear relationship between the terminal voltage and the equivalent internal potential. The magnitude and phase of the fault current of the wind farm group in the steady state will be changed. The coupling relationship is shown in Figure 6.

As shown in Figure 6, the equivalent internal potential E_DFIGi_ and steady-state impedance Z_DFIGi_ of the *i*-th doubly fed wind farm can be obtained from Equation (24) in the fault steady state. Further, the grid admittance matrix Y can be obtained at steady state.
(25)$$Y=\left[\begin{array}{ccccc} Y_{A} & -Y_{\mathrm{AB}} & -Y_{\mathrm{AC}} & 0 & 0\\ -Y_{\mathrm{AB}} & Y_{B} & 0 & -Y_{\mathrm{BD}} & 0\\ -Y_{\mathrm{AC}} & 0 & Y_{C} & 0 & 0\\ 0 & -Y_{\mathrm{BD}} & 0 & Y_{D} & -Y_{\mathrm{DE}}\\ 0 & 0 & 0 & -Y_{\mathrm{DE}} & Y_{E} \end{array}\right]$$

Among them,
(26)$$\left\{\begin{array}{l} Y_{A}=1/Z_{g}+1/Z_{L2}+1/Z_{L1},Y_{\mathrm{AB}}=1/Z_{L2},Y_{\mathrm{AC}}=1/Z_{L1}\\ Y_{B}=1/(Z_{L4}+Z_{\mathrm{DFIG}2})+1/Z_{L2}+1/Z_{L3},Y_{\mathrm{BD}}=1/Z_{L3}\\ Y_{C}=1/Z_{\mathrm{DFIG}1}+1/Z_{L1},Y_{D}=1/(Z_{L5}+Z_{\mathrm{DFIG}3})+1/Z_{L3}+1/Z_{L6}\\ Y_{\mathrm{DE}}=1/Z_{L6},Y_{E}=1/Z_{\mathrm{DFIG}4}+1/Z_{L6} \end{array}\right.$$
where: ***Y_ij_*** is the admittance between nodes, ***Y_g_*** is the system equivalent admittance, and ***Y_DFIGi_*** represents the equivalent admittance of each DFIG.

From Formula (25), the short-circuit current output of each doubly fed wind farm can be obtained as follows:(27)$$\left[\begin{array}{l} I_{g}\\ I_{\mathrm{DFIG}1}\\ I_{\mathrm{DFIG}2}\\ I_{\mathrm{DFIG}3}\\ I_{\mathrm{DFIG}4} \end{array}\right]=Y\left[\begin{array}{l} U_{A}\\ U_{B}\\ U_{C}\\ U_{D}\\ U_{E} \end{array}\right]$$
where *U*_i_ is the voltage of each node; ***I_g_*** and ***I_DFIGi_*** are the short-circuit current provided by the system and each doubly fed wind farm, respectively.

From Formulas (18) and (24), in the power grid voltage drop steady-state moment in the case of short-circuit current of DFIG_i_, the relationship with the voltage of the machine is as follows:(28)$$I_{\mathrm{DFIGi}}=\frac{1}{R_{s}+X}[X_{m}(\frac{u_{s0}}{X_{m}}-\frac{2L_{s}Q_{0}}{3u_{s0}L_{m}})-\frac{2}{3}\frac{jX_{m}L_{s}P_{0}}{L_{m}u_{s0}}-U_{i}]+\frac{X_{m}K_{d}(0.9-U_{i}^{})i_{\mathrm{rN}}}{R_{s}+X}$$

Therefore, this paper uses an iterative correction method to solve the problem. Since the nodal admittance matrix Y is a positive definite symmetric matrix, the iterative process must converge. Therefore, the output current before the fault can be selected as the initial value, and the voltage value of each node and the short-circuit current of each wind farm can be solved. Similarly, the voltages and branch currents of various nodes in the power grid can be obtained through calculations for other types of faults.

Formula (28) is substituted into Formula (27) for solving. Because these formulas are high-order nonlinear systems and cannot be solved directly, the iterative correction solution is carried out using a computer. Since the output current before the fault is selected as the initial value, and the voltage value of each node is solved, as well as the short-circuit current of each wind farm. Similarly, computer iteration can be used to obtain node voltage and branch current in other types of faults.

## 5. Experimental Verification

### 5.1. Experimental Verification of Calculation Method for Short-Circuit Current of DFIG Wind Farm Group

In order to verify the correctness of the short-circuit current calculation method of doubly fed field groups, a model of a doubly fed field group connected to the power grid was established in RTDS.

Taking Figure 3 as an example, the main relevant parameters are as follows: the wind farms DFIG_1_–DFIG_4_ are equipped with 11 sets, 16 sets, 16 sets, and 22 sets of turbines with a rated capacity of 2 MVA turbine.The side of the stator resistance and the leakage inductance are 0.016, and 0.169, respectively, p.u., rotor resistance and leakage inductance are 0.009 and, 0.153, respectively, p.u., and the excitation inductance p.u. is 3.49. The system equivalent impedance is j0.5Ω, the line L1, L2 impedance is (0.194 + j0.487)Ω, L3, L6 impedance is (0.117 + j0.292)Ω, and L4, L5 impedance is (0.019 + j0.049)Ω.

Figure 7 shows the distance point of failure, respectively. The electric nearest and farthest doubly fed DFIG1 and, DFIG4 short-circuit current test results are compared with the model calculation.

Effective values can be extracted from the instantaneous values of the short-circuit current of DFIG1 and DFIG4 measured using the full Fourier algorithm. Figure 7a,b show that the effective values of the short-circuit current of DFIG1 and DFIG4 at the 0.5 s fault instant are 2.25 p.u. and 1.76 p.u., respectively. However, the results calculated in this paper are 2.31 p.u. and 1.72 p.u., and the errors are only 2.7% and 2.4%. In the case of fault steady state, the error is only 1.7% and 1.4%. Therefore, the method proposed in this paper is effective.

In order to make the experiment more accurate, we carried out multiple sets of experimental tests for different fault point conditions under symmetrical working conditions and obtained the results shown in Figure 8, whose errors are all less than 4%.

For further analysis, Figure 9 shows the comparison results of short-circuit current measurement of the positive sequence loop and negative sequence loop of DFIG_1_ and DIFG_4_.

It can be easily seen from Figure 9 that after the occurrence of the 0.5 S fault, the error of the comparison results of the positive sequence short-circuit current of DFIG1 and DFIG4 was only 2% and 2.3%, while the error of negative sequence short-circuit current was only 2.1% and 1.5%. After entering the fault stable state, the error of the positive sequence short-circuit current was only 1.6%,and 1.7%, and the error of the negative sequence short-circuit current was only 1.1% and, 1.6%. Therefore, good results can still be achieved when asymmetric voltage drop conditions occur against the power grid.

In addition, we carried out multiple sets of experimental tests on different fault points for asymmetric working conditions, and obtained the results shown in Figure 10. The errors were all less than 4%, which verifies that the calculation method in this paper also has considerable accuracy for asymmetric working conditions.

### 5.2. Experimental Verification of Fault Analysis Method for Doubly Fed Wind Farm Group

Figure 3 is taken as an example to verify the short-circuit fault on the connected grid side of the doubly fed field group. Table 1 and Table 2 show the positive and negative sequence short-circuit current comparison of the branches corresponding to DFIG1-4 when point A is in symmetric drop and asymmetric drop, respectively. It can be seen that the short-circuit current amplitude error is only 2.5% and the phase angle error is only 10.43% under the symmetrical drop condition. This is much smaller than the calculation error of the traditional calculation method using the equivalent synchronous generator.

## 6. Conclusions

Because the traditional method is simple and equivalent to the synchronous generator, the fault analysis method is not applicable to the wind power generation mode of cluster development and centralized grid connection in our country, the influence of the LVRT control strategy on short-circuit current calculation is comprehensively considered in this paper, and the RTDS model is established. The coupling relationship between the wind farms and the voltage of the grid node is further given, finally, the grid fault analysis method suitable for doubly fed wind farm group access is proposed.

(1)The control strategy adopted during the fault period will have a great influence on the short-circuit current characteristics of doubly fed wind turbines. China’s new wind power grid-connected standard puts forward requirements for the output reactive support current of doubly fed wind turbines during the fault period, and it is necessary to calculate the short-circuit current of doubly fed wind turbines and its influence.(2)There is a strong coupling relationship between the short-circuit current of each wind farm in the wind farm group and the voltage of the grid node. The coupling relationship must be considered to obtain a more accurate output short-circuit current of each wind farm.

## Figures and Tables

**Figure 1 sensors-23-08372-f001:**
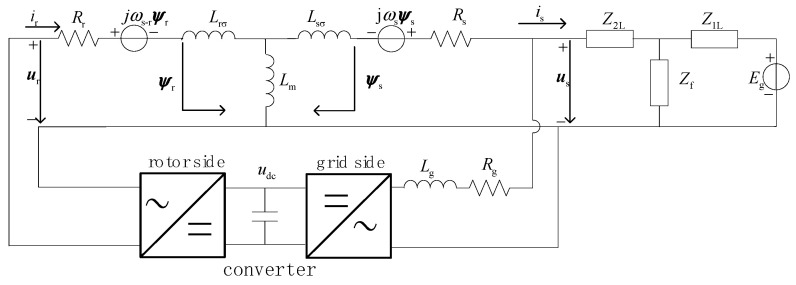
DFIG equivalent circuit under the condition of power grid voltage sags.

**Figure 2 sensors-23-08372-f002:**
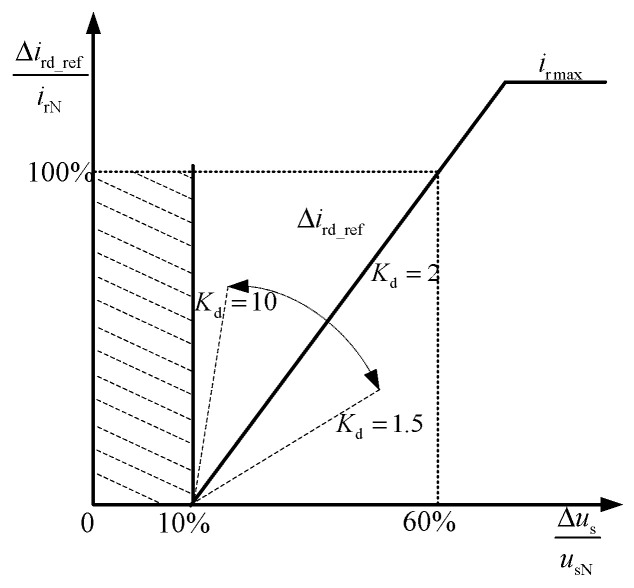
Reactive current reference value and the relationship between the degree of voltage sags.

**Figure 3 sensors-23-08372-f003:**
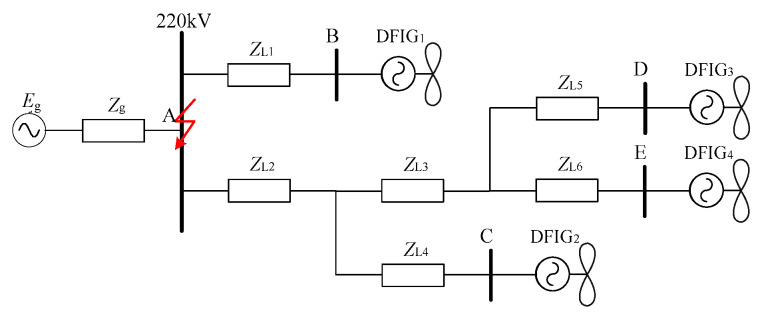
Diagram of power grid with doubly fed wind farm groups.

**Figure 4 sensors-23-08372-f004:**
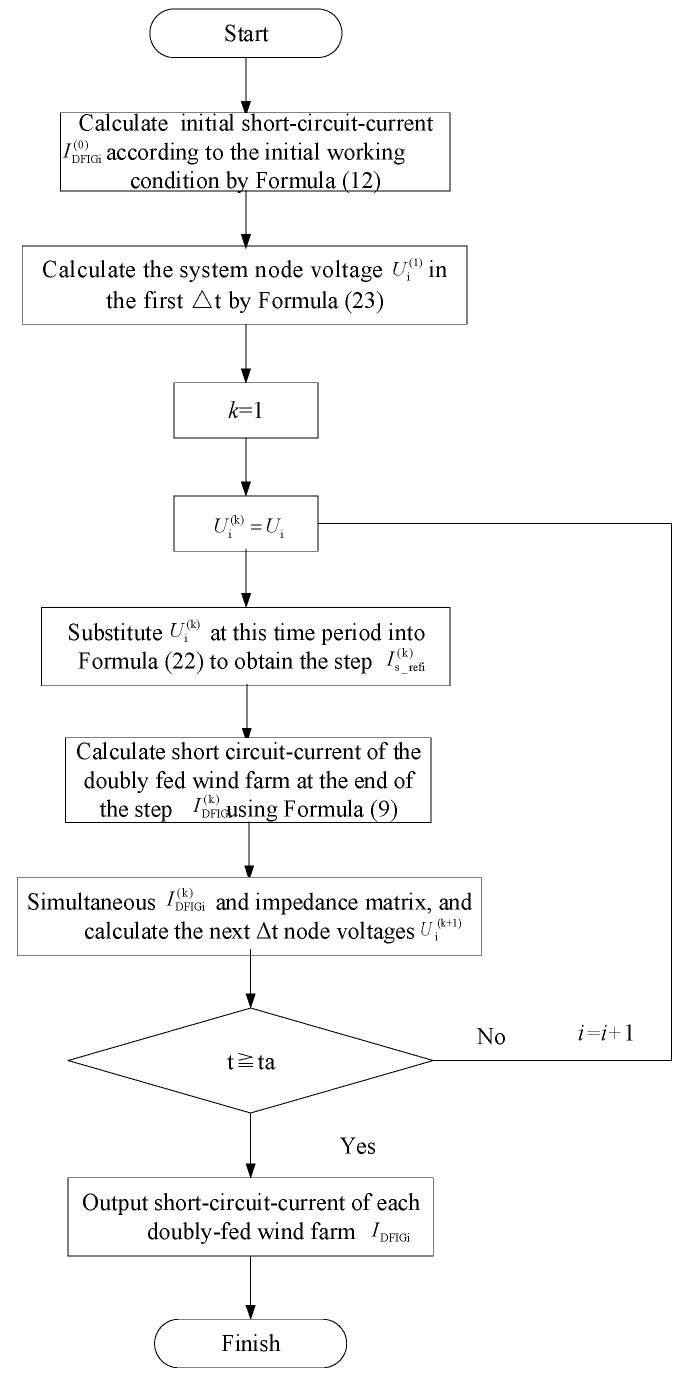
Flow chart of doubly fed wind farm group short-circuit-current calculation method.

**Figure 5 sensors-23-08372-f005:**
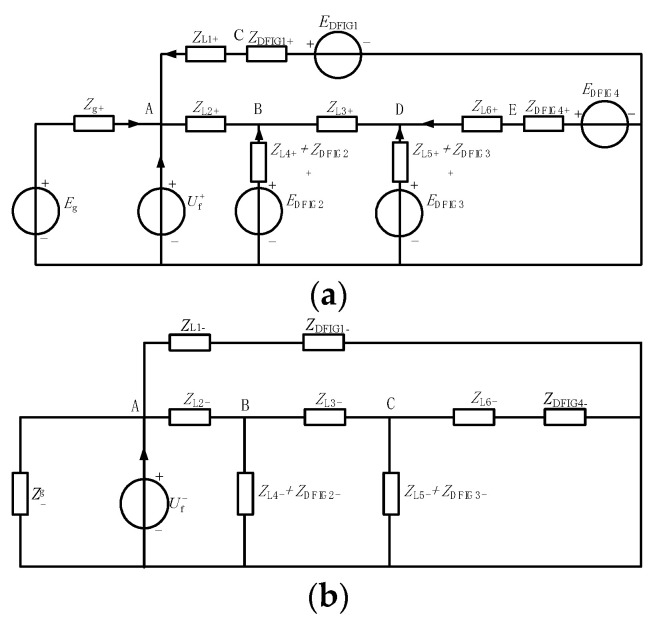
Composite order network diagram of power grid with DFIG field group access. (**a**) Positive sequence network after doubly fed wind farm group is connected to power grid; (**b**) negative sequence network after doubly fed wind farm group is connected to power grid.

**Figure 6 sensors-23-08372-f006:**
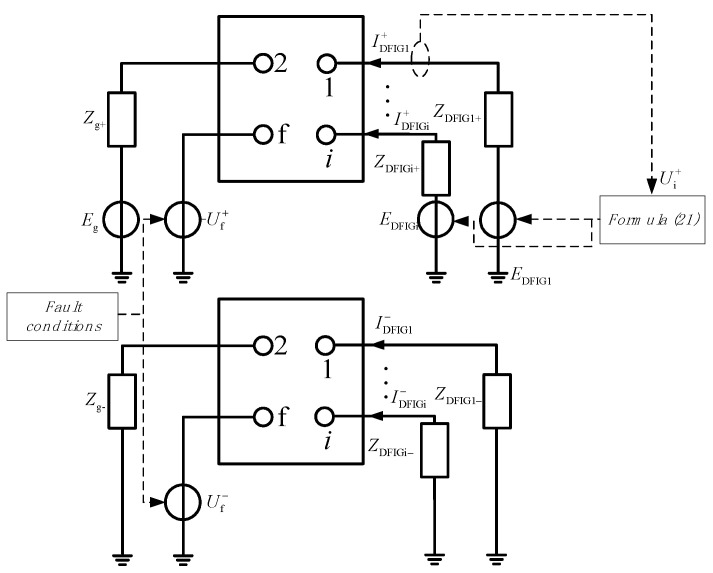
Coupling relation between doubly fed wind farm group and network.

**Figure 7 sensors-23-08372-f007:**
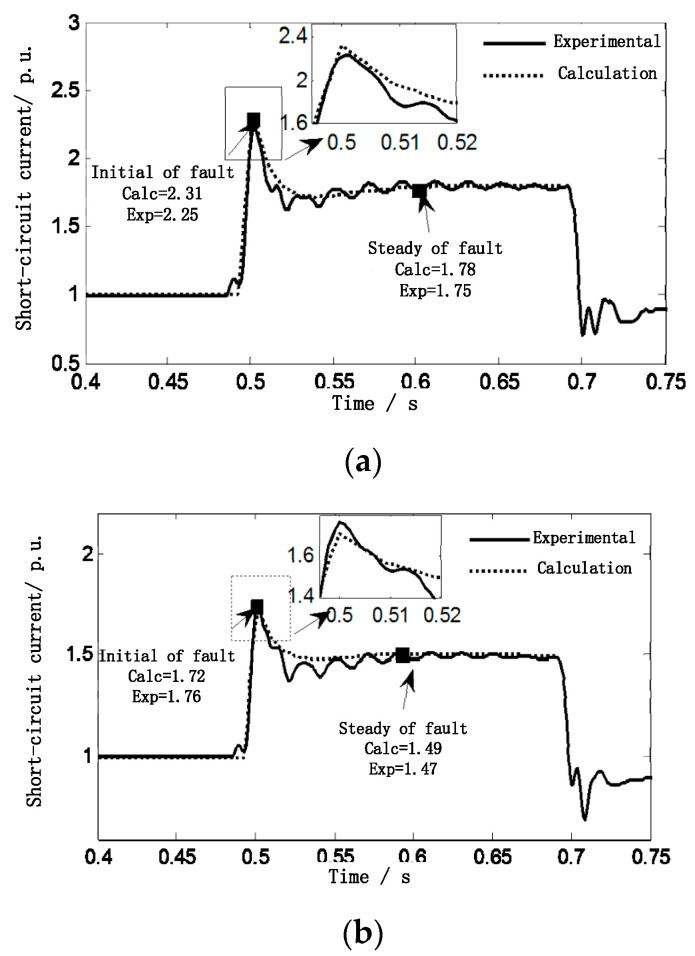
A point voltage three-phase short-circuit fault comparison of short-circuit current calculation and measured value of each wind farm. (**a**) Comparison of DFIG_1_ short-circuit current calculated results and experimental results. (**b**) Comparison between calculated and measured DFIG_4_ short-circuit current of doubly fed wind farm.

**Figure 8 sensors-23-08372-f008:**
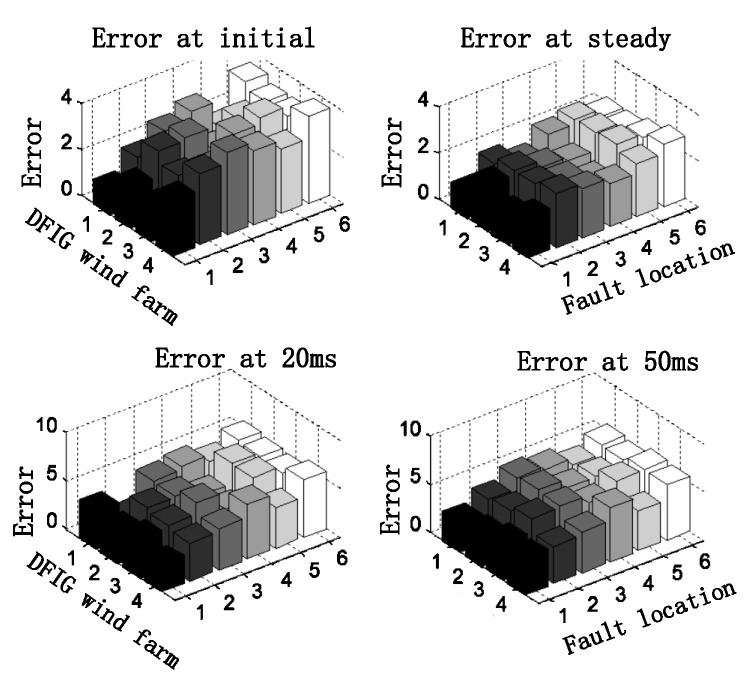
Error between the calculated results of the short-circuit current model and the measured results in three-phase short-circuit.

**Figure 9 sensors-23-08372-f009:**
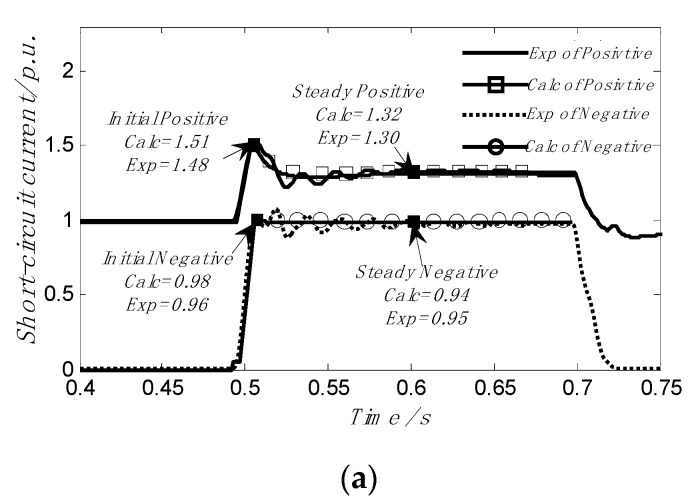
The measured short-circuit current is compared with the calculated results when the two-phase short circuit of the grid voltage occurs at terminal A. (**a**) Comparison of DFIG_1_ experimental and calculated results for positive and negative sequence short-circuit current; (**b**) comparison of DFIG_4_ experimental and calculated results for positive and negative sequence short-circuit current.

**Figure 10 sensors-23-08372-f010:**
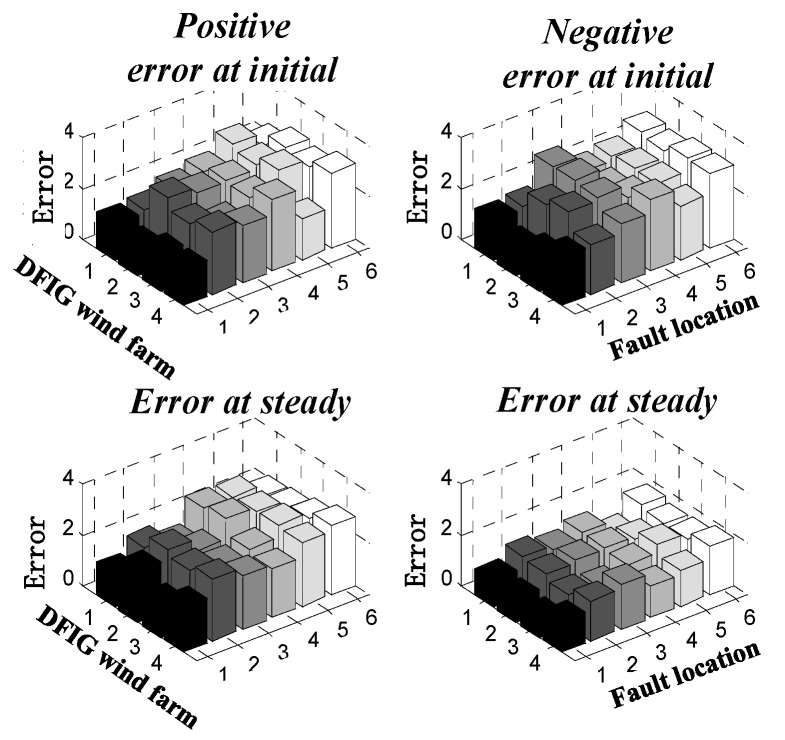
Calculation error between calculation results and experimental results for two-phase short circuit.

**Table 1 sensors-23-08372-t001:** The calculated values of the model were compared with the experimental test values in the case of a three-phase short circuit.

	Branch Current *I*_L1_		Branch Current *I*_L4_		Branch Current *I*_L5_		Branch Current *I*_L6_	
	Amplitude /kA	Phase Angle/(°)	Amplitude /kA	Phase Angle/(°)	Amplitude /kA	Phase Angle/(°)	Amplitude /kA	Phase Angle/(°)
Before fault	0.057	23.09	0.084	26.37	0.084	27.96	0.115	18.94
Experimental value	0.091	−28.81	0.126	−24.7	0.117	−22.41	0.202	−31.36
Calculation results	0.093	−34.57	0.129	−28.43	0.12	−25.46	0.207	−41.79
Traditional method	0.189	−41.31	0.253	−38.23	0.243	−37.81	0.415	−46.59

**Table 2 sensors-23-08372-t002:** The model calculated values were compared with the experimental test values when the two phases of AB were short-circuited.

	Branch Current *I*_L1_				Branch Current *I*_L4_				Branch Current *I*_L5_				Branch Current *I*_L6_			
	Positive Seq		Negative Seq		Positive Seq		Negative Seq		Positive Seq		Negative Seq		Positive Seq		Negative Seq	
	Amp /kA	Ang /(°)	Amp /kA	Ang /(°)	Amp /kA	Ang /(°)	Amp /kA	Ang /(°)	Amp /kA	Ang /(°)	Amp /kA	Ang /(°)	Amp /kA	Ang /(°)	Amp /kA	Ang /(°)
Before fault	0.057	23.09	0	0	0.084	26.37	0	0	0.084	27.96	0	0	0.115	18.94	0	0
Experiments	0.073	−21.61	0.042	−23.06	0.104	−17.3	0.057	−23.48	0.101	−15.08	0.051	−23.37	0.155	−24.93	0.107	−23.5
Calculation	0.075	−26.78	0.041	−23.82	0.106	−24.09	0.056	−24.6	0.104	−22.53	0.049	−21.29	0.16	−28.215	0.105	−25. 7
Traditional method	0.141	−37.96	0.062	−19.42	0.193	−33.42	0.081	−20.26	0.185	−30.69	0.075	−20.85	0.294	−41.25	0.134	−21.41

## Data Availability

The original contributions presented in this study are included in the article. Further inquiries can be directed to the corresponding authors.

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
