# Peer review of "A New Short-Circuit Current Calculation and Fault Analysis Method Suitable for Doubly Fed Wind Farm Groups"

_sensors, 2023, doi:10.3390/s23208372_

Round 1

Reviewer 1 Report

See the attachment.

See the attachment.

Reviewer 2 Report

1.The derivation of Equation 4 to  5 is confusing.

2.The short circuit current calculation method is too cumbersome to express, and may be considered to be streamlined.

3.What specifically is the traditional methods?

none.

Reviewer 3 Report

I am satisfying with current manuscript.

The following issue can be addressed, that can make this manuscript better

1. In Fig. 1, please specify the number for the parameters of the equivalent circuit.

2. In Fig. 3, pleae specify the ABCDE group, what are the purpose for those area?

3. In Fig. 4, the flow for doubly fed wind is not clear, please give the comment in the paragraph as well.

Reviewer 4 Report

The transient model of doubly-fed wind turbine is proposed, and the influence mechanism of control strategy on short-circuit current under power grid fault is analyzed in this paper. The following problems still exist, and the author is suggested to revise them carefully.

1. Please explain Figure 3 further.

2. Generally speaking, when asymmetric faults occur in the power grid, it is difficult to obtain satisfactory performance if the single synchronous coordinate system PLL is still used before the fault. Please indicate which PLL can be used to accurately obtain the positive sequence voltage phase and amplitude of the DFIG converter in the case of fault.

3. Please add some references related to short-circuit current calculation of doubly-fed field group.

4. Why can't we use the traditional short-circuit current method? Please explain the advantages of the proposed short-circuit current calculation method compared to the traditional fault analysis method.

5.Since the doubly-fed fan is connected to the grid in the form of field group, how to deal with the coupling effect between them in the calculation of short circuit?

6. It is suggested to add some high quality and relevant research,and it is also recommended that the author add some references of MDPI.

[1] Kou, L.; Li, Y.; Zhang, F.; Gong, X.; Hu, Y.; Yuan, Q.; Ke, W. Review on Monitoring, Operation and Maintenance of Smart Offshore Wind Farms. Sensors 2022, 22, 2822. https://doi.org/10.3390/s22082822

[2] Bi, X.; Yang, J.; Yang, S. LCA-Based Regional Distribution and Transference of Carbon Emissions from Wind Farms in China. Energies 2022, 15, 198. https://doi.org/10.3390/en15010198

Round 2

Reviewer 1 Report

No further comment.

Minor editing of English language required